## SCIENCE FORUM

# The Human Cell Atlas

**Abstract** The recent advent of methods for high-throughput single-cell molecular profiling has catalyzed a growing sense in the scientific community that the time is ripe to complete the 150-year-old effort to identify all cell types in the human body. The Human Cell Atlas Project is an international collaborative effort that aims to define all human cell types in terms of distinctive molecular profiles (such as gene expression profiles) and to connect this information with classical cellular descriptions (such as location and morphology). An open comprehensive reference map of the molecular state of cells in healthy human tissues would propel the systematic study of physiological states, developmental trajectories, regulatory circuitry and interactions of cells, and also provide a framework for understanding cellular dysregulation in human disease. Here we describe the idea, its potential utility, early proofs-of-concept, and some design considerations for the Human Cell Atlas, including a commitment to open data, code, and community.
DOI: https://doi.org/10.7554/eLife.27041.001

AVIV REGEV*, SARAH A  TEICHMANN*, ERIC S  LANDER*, IDO AMIT, CHRISTOPHE BENOIST, EWAN BIRNEY, BERND BODENMILLER, PETER CAMPBELL, PIERO CARNINCI, MENNA CLATWORTHY, HANS CLEVERS, BART DEPLANCKE, IAN DUNHAM, JAMES EBERWINE, ROLAND EILS, WOLFGANG ENARD, ANDREW FARMER, LARS FUGGER, BERTHOLD GÖTTGENS, NIR HACOHEN, MUZLIFAH HANIFFA, MARTIN HEMBERG, SEUNG KIM, PAUL KLENERMAN, ARNOLD KRIEGSTEIN, ED LEIN, STEN LINNARSSON, EMMA LUNDBERG, JOAKIM LUNDEBERG, PARTHA MAJUMDER, JOHN C  MARIONI, MIRIAM MERAD, MUSA MHLANGA, MARTIJN NAWIJN, MIHAI NETEA, GARRY NOLAN, DANA PE'ER, ANTHONY PHILLIPAKIS, CHRIS P  PONTING, STEPHEN QUAKE, WOLF REIK, ORIT ROZENBLATT-ROSEN, JOSHUA SANES, RAHUL SATIJA, TON N SCHUMACHER, ALEX SHALEK, EHUD SHAPIRO, PADMANEE SHARMA, JAY W SHIN, OLIVER STEGLE, MICHAEL STRATTON, MICHAEL J T  STUBBINGTON, FABIAN J  THEIS, MATTHIAS UHLEN, ALEXANDER VAN OUDENAARDEN, ALLON WAGNER, FIONA WATT, JONATHAN WEISSMAN, BARBARA WOLD, RAMNIK XAVIER, NIR YOSEF AND HUMAN CELL ATLAS MEETING PARTICIPANTS

*For correspondence: aregev@ broadinstitute.org (AR); st9@ sanger.ac.uk (SAT); eric@ broadinstitute.org (ESL)

## Introduction

The cell is the fundamental unit of living organisms. Hooke reported the discovery of cells in plants in 1665 (*Hooke, 1665*) and named them for their resemblance to the cells inhabited by monks, but it took nearly two centuries for biologists to appreciate their central role in biology. Between 1838 and 1855, Schleiden, Schwann, Remak, Virchow and others crystalized an elegant Cell Theory (*Harris, 2000*), stating that all organisms are composed of one or more cells; that cells are the basic unit of structure and function in life; and that all cells are derived from pre-existing cells (*Mazzarello, 1999*; *Figure 1*).

To study human biology, we must know our cells. Human physiology emerges from normal cellular functions and intercellular interactions. Human disease entails the disruption of these processes and may involve aberrant cell types and states, as seen in cancer. Genotypes give rise to organismal phenotypes through the intermediate of cells, because cells are the basic functional units, each regulating their own program of gene expression. Therefore, genetic variants that contribute to disease typically manifest their action through impact in a particular cell types: for example, genetic variants in the *IL23R* locus increase risk of autoimmune diseases by

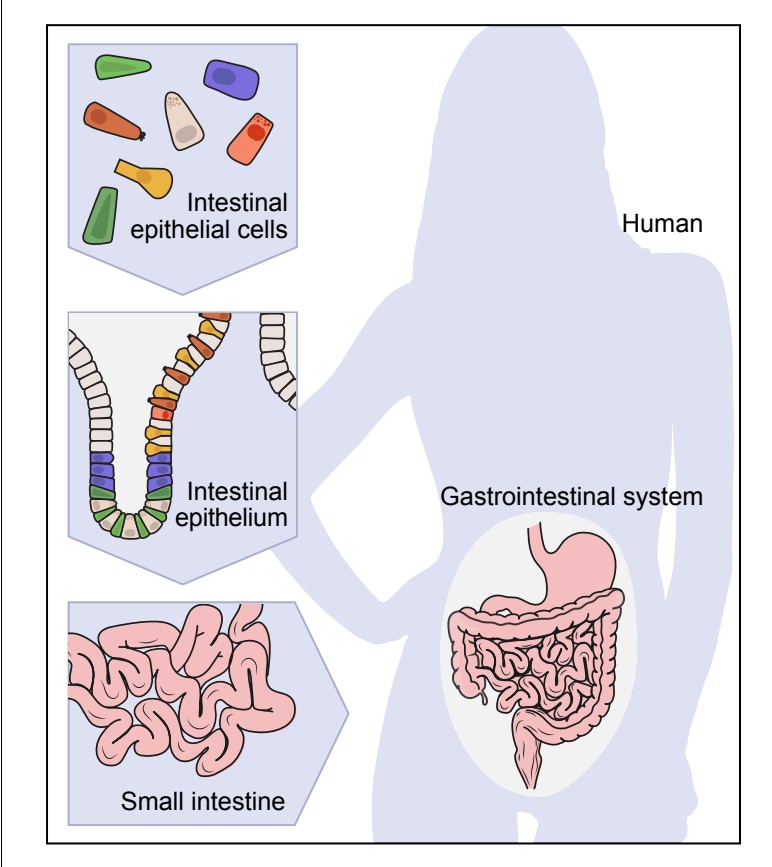

**Figure 1.** A hierarchical view of human anatomy. A graphical depiction of the anatomical hierarchy from organs (such as the gut), to tissues (such as the epithelium in the crypt in the small intestine), to their constituent cells (such as epithelial, immune, stromal and neural cells).

DOI: https://doi.org/10.7554/eLife.27041.002

discovered by Camillo Golgi to show that the brain is composed of distinct neuronal cells, rather than a continuous syncytium, with stunningly diverse architectures found in specific anatomical regions (*Ramón y Cajal, 1995*); the pair shared the 1906 Nobel Prize in Physiology or Medicine for their work.

Starting in the 1930s, electron microscopy provided up to 5000-fold higher resolution, making it possible to discover and distinguish cells based on finer structural features. Immunohistochemistry, pioneered in the 1940s (*Arthur, 2016*) and accelerated by the advent of monoclonal antibodies (*Köhler and Milstein, 1975*) and Fluorescence-Activated Cell Sorting (FACS; *Dittrich and Göhde, 1971*; *Fulwyler, 1965*) in the 1970s, made it possible to detect the presence and levels of specific proteins. This revealed that morphologically indistinguishable cells can vary dramatically at the molecular level and led to exceptionally fine classification systems, for example, of hematopoietic cells, based on cell-surface markers. In the 1980s, Fluorescence *in situ* Hybridization (FISH; *Langer-Safer et al., 1982*) enhanced the ability to characterize cells by detecting specific DNA loci and RNA transcripts. Along the way, studies showed that distinct molecular phenotypes typically signify distinct functionalities. Through these remarkable efforts, biologists have achieved an impressive understanding of specific systems, such as the hematopoietic and immune systems (*Chao et al., 2008*; *Jojic et al., 2013*; *Kim and Lanier, 2013*) or the neurons in the retina (*Sanes and Masland, 2015*).

Despite this progress, our knowledge of cell types remains incomplete. Moreover, current classifications are based on different criteria, such as morphology, molecules and function, which have not always been related to each other. In addition, molecular classification of cells has largely been ad hoc – based on markers discovered by accident or chosen for convenience – rather than systematic and comprehensive. Even less is known about cell states and their relationships during development: the full lineage tree of cells from the single-cell zygote to the adult is only known for the nematode *C. elegans*, which is transparent and has just ~1000 cells.

At a conceptual level, one challenge is that we lack a rigorous definition of what we mean by the intuitive terms 'cell type' and 'cell state'.

altering the function of dendritic cells and T-cells (*Duerr et al., 2006*), and DMD mutations cause muscular dystrophy through specific effects in skeletal muscle cells (*Murray et al., 1982*).

For more than 150 years, biologists have sought to characterize and classify cells into distinct types based on increasingly detailed descriptions of their properties, including their shape, their location and relationship to other cells within tissues, their biological function, and, more recently, their molecular components. At every step, efforts to catalog cells have been driven by advances in technology. Improvements in light microscopy were obviously critical. So too was the invention of synthetic dyes by chemists (*Nagel, 1981*), which biologists rapidly found stained cellular components in different ways (*Stahnisch, 2015*). In pioneering work beginning in 1887, Santiago Ramón y Cajal applied a remarkable staining process

Cell type often implies a notion of persistence (*e.g.*, being a hepatic stellate cell or a cerebellar Purkinje cell), while cell state often refers to more transient properties (*e.g.*, being in the G1 phase of the cell cycle or experiencing nutrient deprivation). But, the boundaries between these concepts can be blurred, because cells change over time in ways that are far from fully understood. Ultimately, data-driven approaches will likely refine our concepts.

The desirability of having much deeper knowledge about cells has been well recognized for a long time (*Brenner, 2010*; *Eberwine et al., 1992*; *Shapiro, 2010*; *Van Gelder et al., 1990*). However, only in the past few years has it begun to seem feasible to undertake the kind of systematic, high-resolution characterization of human cells necessary to create a systematic cell atlas.

The key has been the recent ability to apply genomic profiling approaches to single cells. By 'genomic approaches' we mean methods for large-scale profiling of the genome and its products, including DNA sequence, chromatin architecture, RNA transcripts, proteins, and metabolites (*Lander, 1996*). It has long been appreciated that such methods provide rich and comprehensive descriptions of biological processes. Historically, however, they could only be applied to bulk tissue samples comprised of an ensemble of many cells, providing average genomic measures for a sample, but masking their differences across cells. The result is as unsatisfying as trying to understand New York, London or Mumbai based on the average properties of their inhabitants.

The first single-cell genomic characterization method to become feasible at large-scale is trancriptome analysis by single cell RNA-Seq (*Box 1*; *Hashimshony et al., 2012*; *Jaitin et al., 2014*; *Picelli et al., 2013*; *Ramsköld et al., 2012*; *Shalek et al., 2013*). Initial efforts first used microarrays and then RNA-seq to profile RNA from small numbers of single cells, which were obtained either by manual picking from in situ fixed tissue, using flow-sorting or, later on, with microfluidic devices, adapted from devices developed initially for qPCR-based approaches (*Crino et al., 1996*; *Dalerba et al., 2011*; *Marcus et al., 2006*; *Miyashiro et al., 1994*; *Zhong et al., 2008*). Now, massively parallel assays can process tens and hundreds of thousands of single cells simultaneously to measure their transcriptional profiles at rapidly decreasing costs (*Klein et al., 2015*; *Macosko et al., 2015*; *Shekhar et al., 2016*) with increasing

accuracy and sensitivity (*Svensson et al., 2017*; *Ziegenhain et al., 2017*). In some cases, it is even possible to register these sorted cells to their spatial positions in images (*Vickovic et al., 2016*). Single-cell RNA sequencing (scRNA-seq) is rapidly becoming widely disseminated.

Following this initial wave of technologies are many additional methods at various stages of development and high-throughput implementation. Techniques are being developed to assay: *in situ* gene expression in tissues at single-cell and even sub-cellular resolution (*Chen et al., 2015b*; *Ke et al., 2013*; *Lee et al., 2014*; *Lubeck et al., 2014*; *Shah et al., 2016*; *Ståhl et al., 2016*); the distribution of scores of proteins at cellular or sub-cellular resolution (*Angelo et al., 2014*; *Chen et al., 2015a*; *Giesen et al., 2014*; *Hama et al., 2011*; *Susaki et al., 2014*; *Yang et al., 2014*); various aspects of chromatin state (*Buenrostro et al., 2015*; *Cusanovich et al., 2015*; *Farlik et al., 2015*; *Guo et al., 2013*; *Lorthongpanich et al., 2013*; *Mooijman et al., 2016*; *Rotem et al., 2015a*; *Rotem et al., 2015b*; *Smallwood et al., 2014*); and DNA mutations to allow precise reconstruction of cell lineages (*Behjati et al., 2014*; *Biezuner et al., 2016*; *Shapiro et al., 2013*; *Taylor et al., 2003*; *Teixeira et al., 2013*). Various groups are also developing single-cell multi-omic methods to simultaneously measure several types of molecular profiles in the same cell (*Albayrak et al., 2016*; *Angermueller et al., 2016*; *Behjati et al., 2014*; *Darmanis et al., 2016*; *Dey et al., 2015*; *Frei et al., 2016*; *Genshaft et al., 2016*; *Macaulay et al., 2015*).

As a result, there is a growing sense in the scientific community that the time is now right for a project to complete the Human Cell Atlas that pioneering histologists began 150 years ago. Various discussions have taken place in a number of settings over the past two years, culminating in an international meeting in London in October 2016. In addition, several pilot efforts are already underway or in planning – for example, related to brain cells and immune cells. Prompted by such efforts, funding agencies, including the NIH, have sought information from the scientific community about the notion of creating cell or tissue atlases.

The goal of this article is to engage the wider scientific community in this conversation. Although the timing is driven by technologies that have recently appeared or are expected to mature in the near-future, the project itself is fundamentally an intellectual endeavor. We

## Box 1: Key experimental methods for single-cell genomics

Over the past several years, powerful approaches have emerged that make it possible to measure molecular profiles and signatures at single-cell resolution. The field remains very active, with new methods being rapidly developed and existing ones improved.

*Single-cell RNA-Seq* (scRNA-seq) refers to a class of methods for profiling the transcriptome of individual cells. Some may take a census of mRNA species by focusing on 3'- or 5'-ends (*Islam et al., 2014*; *Macosko et al., 2015*), while others assess mRNA structure and splicing by collecting near-full-length sequence (*Hashimshony et al., 2012*; *Ramsköld et al., 2012*). Strategies for single-cell isolation span manual cell picking, initially used in microarray studies (*Eberwine et al., 1992*; *Van Gelder et al., 1990*), FACS-based sorting into multi-well plates (*Ramsköld et al., 2012*; *Shalek et al., 2013*), microfluidic devices (*Shalek et al., 2014*; *Treutlein et al., 2014*), and, most recently, droplet-based (*Klein et al., 2015*; *Macosko et al., 2015*) and microwell-based (*Fan et al., 2015*; *Yuan and Sims, 2016*) approaches. The droplet and microwell approaches, which are currently coupled to 3'-end counting, have the largest throughput, allowing rapid processing of tens of thousands of cells simultaneously in a single sample. scRNA-seq is typically applied to freshly dissociated tissue, but emerging protocols use fixed cells (*Nichterwitz et al., 2016*; *Thomsen et al., 2016*) or nuclei isolated from frozen or lightly fixed tissue (*Habib et al., 2016b*; *Lake et al., 2016*). Applications to fixed or frozen samples would simplify the process flow for scRNA-seq, as well as open the possibility of using archival material. Power analyses provides a framework for comparing the sensitivity and accuracy of these approaches (*Svensson et al., 2017*; *Ziegenhain et al., 2017*). Finally, there has been progress in scRNA-Seq with RNA isolated from live cells in their natural microenvironment using transcriptome in vivo analysis (*Lovatt et al., 2014*).

*Mass cytometry (CyTOF)* and related methods allow multiplexed measurement of proteins based on antibodies barcoded with heavy metals (*Bendall et al., 2014*; *Levine et al., 2015*). In contrast to comprehensive profiles, these methods invglve pre-defined signatures and require an appropriate antibody for each target, but they can process many millions of cells for a very low cost per cell. They are applied to fixed cells. Recently, the approach has been extended to the measurement of RNA signatures through multiplex hybridization of nucleic-acid probes tagged with heavy metals (*Frei et al., 2016*).

*Single-cell genome and epigenome sequencing* characterizes the cellular genome. Genomic methods aim either to characterize the whole genome or capture specific pre-defined regions (*Gao et al., 2016*). Epigenomic methods may capture regions based on distinctive histone modifications (single-cell ChIP-Seq; *Rotem et al., 2015a*), accessibility (single-cell ATAC-Seq; *Buenrostro et al., 2015*; *Cusanovich et al., 2015*), or likewise characterize DNA methylation patterns (single-cell DNAme-Seq; *Farlik et al., 2015*; *Guo et al., 2013*; *Mooijman et al., 2016*; *Smallwood et al., 2014*) or 3D organization (single-cell Hi-C; *Nagano et al., 2013*; *Ramani et al., 2017*). Combinatorial barcoding strategies have been used to capture measures of accessibility and 3D organization in tens of thousands of single cells (*Cusanovich et al., 2015*; *Ramani et al., 2017*). Single cell epigenomics methods are usually applied to nuclei, and can thus use frozen or certain fixed samples. Some methods, such as single-cell DNA sequencing, are currently applied to relatively few cells, due to the size of the genome and the sequencing depth required. Other methods, such as single-cell analysis of chromatin organization (by either single-cell ATAC-Seq; *Buenrostro et al., 2015*; *Cusanovich et al., 2015*) or single-cell ChIP-Seq (*Rotem et al., 2015a*), currently yield rather sparse data, which presents analytic challenges and benefits from large numbers of profiled cells. Computational analyses have begun to address these issues by pooling of signal across cells and across genomic regions or loci (*Buenrostro et al., 2015*; *Rotem et al., 2015a*) and by imputation (*Angermueller et al., 2016*).

*Single-cell multi-omics* techniques aim to collect two or more types of data (transcriptomic, genomic, epigenomic, and proteomic) from the same single cell. Recent studies have simultaneously profiled the transcriptome together with either the genome (*Angermueller et al., 2016*; *Dey et al., 2015*; *Macaulay et al., 2015*), the epigenome (*Angermueller et al., 2016*), or protein signatures (*Albayrak et al., 2016*; *Darmanis et al., 2016*; *Frei et al., 2016*; *Genshaft et al.,*

2016). Efforts to combine three and more approaches are underway (*Cheow et al., 2016*). Multi-omic methods could help fill in causal chains from genetic variation to regulatory mechanisms and phenotypic outcome in health and in disease, especially cancer.

*Multiplex in situ analysis and other spatial techniques* aim to detect a limited number of nucleic acids and/or proteins *in situ* in tissue samples – by hybridization (for RNA), antibody staining (for proteins), sequencing (for nucleic acids), or other tagging strategies. These *in situ* results can then be used to map massive amounts of single-cell genomic information from dissociated cells onto the tissue samples providing important clues about spatial relationships and cell-cell communication. Some strategies for RNA detection, such as MERFISH (*Chen et al., 2015b*; *Moffitt et al., 2016b*) or Seq-FISH (*Shah et al., 2016*), combine multiplex hybridization with microscopy-based quantification to assess distributions at both the cellular and subcellular level; other early studies have performed *in situ* transcription (*Tecott et al., 1988*), followed by direct manual harvesting of cDNA from individual cells (*Crino et al., 1996*; *Tecott et al., 1988*). Some approaches for protein detection, such as Imaging Mass Cytometry (*Giesen et al., 2014*) and Mass Ion Bean Imaging (*Angelo et al., 2014*), involve staining a tissue specimen with antibodies, each labeled with a barcode of heavy metals, and rastering across the sample to measure the proteins in each 'pixel'. This technique permits the reconstruction of remarkably rich images. Finally, more recent studies have performed RNA-seq *in situ* in cells and in preserved tissue sections (*Ke et al., 2013*; *Lee et al., 2014*). Many *in situ* methods can benefit from tissue clearing and/or expansion to improve detection and spatial resolution (*Chen et al., 2015a*; *Chen et al., 2016a*; *Moffitt et al., 2016a*; *Yang et al., 2014*). The complexity and accuracy of these methods continues to improve with advances in sample handling, chemistry and imaging. Various methods are also used, for example, to measure transcriptomes *in situ* with barcoded arrays (*Ståhl et al., 2016*).

*Cell lineage determination* Because mammals are not transparent and have many billions of cells, it is not currently possible to directly observe the fate of cells by microscopy. Various alternative approaches have been developed (*Kretzschmar and Watt, 2012*). In mice, cells can be genetically marked with different colors (*Barker et al., 2007*) or DNA barcodes (*Lu et al., 2011*; *Naik et al., 2013*; *Perié and Duffy, 2016*), and their offspring traced during development. Recent work has used iterative CRISPR-based genome editing to generate random genetic scars in the fetal genome and use them to reconstruct lineages in the adult animal (*McKenna et al., 2016*). In humans, where such methods cannot be applied, human cell lineages can be monitored experimentally *in vitro*, or by transplantation of human cells to immunosuppressed mice (*Morton and Houghton, 2007*; *O'Brien et al., 2007*; *Richmond and Su, 2008*), or can be inferred from *in vivo* samples by measuring the DNA differences between individual sampled cells, arising from random mutations during cell division, and using the genetic distances to construct cellular phylogenies, or lineages (*Behjati et al., 2014*; *Shapiro et al., 2013*).

DOI: https://doi.org/10.7554/eLife.27041.003

therefore articulate the concept of a cell atlas and explore its potential utility for biology and medicine. We discuss how an atlas can lead to new understanding of histology, development, physiology, pathology, and intra- and inter-cellular regulation, and enhance our ability to predict the impact of perturbations on cells. It will also yield molecular tools with applications in both research and clinical practice.

As discussed below, a Human Cell Atlas Project would be a shared international effort involving diverse scientific communities. More details are available in the Human Cell Atlas White Paper (https://www.humancellatlas.org/files/HCA_WhitePaper_18Oct2017.pdf): the first version of this 'living document', which will updated on a regular basis, was released on October 18, 2017.

## What is the Human Cell Atlas, and what could we learn from it?

At its most basic level, the Human Cell Atlas must include a comprehensive reference catalog of all human cells based on their stable properties and transient features, as well as their locations and abundances. Yet, an atlas is more than just a catalog: it is a *map* that aims to show the *relationships* among its elements. By doing so, it can sometimes reveal fundamental processes –

## Box 2: On Exactitude in Science. Jorge Luis Borges (1946)

".. . In that Empire, the Art of Cartography attained such Perfection that the map of a single Province occupied the entirety of a City, and the map of the Empire, the entirety of a Province. In time, those Unconscionable Maps no longer satisfied, and the Cartographers Guilds struck a Map of the Empire whose size was that of the Empire, and which coincided point for point with it. The following Generations, who were not so fond of the Study of Cartography as their Forebears had been, saw that that vast map was Useless, and not without some Pitilessness was it, that they delivered it up to the Inclemencies of Sun and Winters. In the Deserts of the West, still today, there are Tattered Ruins of that Map, inhabited by Animals and Beggars; in all the Land there is no other Relic of the Disciplines of Geography."

Purportedly from Suárez Miranda, Travels of Prudent Men, Book Four, Ch. XLV, Lérida, 1658.

DOI: https://doi.org/10.7554/eLife.27041.004

akin to how the atlas of Earth suggested continental drift through the correspondence of coastlines.

To be useful, an atlas must also be an abstraction, comprehensively representing certain features, while ignoring others. The writer Jorge Luis Borges – a master at capturing the tension between grandeur and grandiosity – distilled this challenge in his one-paragraph story, "*On Exactitude in Science*", about an empire enamored with science of cartography (*Box 2*; *Borges and Hurley, 2004*). Over time, the cartographers' map of the realm grew more and more elaborate, and hence bigger, until – *expandio ad absurdum* – the map reached the size of the entire empire itself and became useless.

Moreover, an atlas must provide a system of coordinates on which one can represent and harmonize concepts at many levels (geopolitical borders, topography, roads, climate, restaurants, and even dynamic traffic patterns). Features can be viewed at any level of magnification, and high-dimensional information collapsed into simpler views.

So, a key question is how a Human Cell Atlas should abstract key features, provide coordinates, and show relationships. A natural solution would be to describe each human cell by a defined set of molecular markers. For example, one might describe each cell by the expression level of each of the ~20,000 human protein-coding genes: that is, each cell would be represented as a point in ~20,000-dimensional space.

Of course, the set of markers could be expanded to include the expression levels of non-coding genes, the levels of the alternatively spliced forms of each transcript, the chromatin state of every promoter and enhancer, and the levels of each protein or each post-translationally modified form of each protein. The optimal amount and type of information to collect will emerge based on a balance of technological feasibility and the biological insight provided by each layer (*Corces et al., 2016*; *Lorthongpanich et al., 2013*; *Paul et al., 2015*). For specific applications, it will be useful to employ reduced representations. Solely for concreteness, we will largely refer below to the 20,000-dimensional space of gene expression, which can already be assayed at high-throughput.

The Atlas should have additional coordinates or annotations to represent histological and anatomical information (*e.g.*, a cell's location, morphology, or tissue context), temporal information (*e.g.*, the age of the individual or time since an exposure), and disease status. Such information is essential for harmonizing results based on molecular profiles with rich knowledge about cell biology, histology and function. How best to capture and represent this information requires serious attention.

In some respects, the Human Cell Atlas Project (whose fundamental unit is a cell) is analogous to the Human Genome Project (whose fundamental unit is a gene). Both are ambitious efforts to create 'Periodic Tables' for biology

that comprehensively enumerate the two key 'atomic' units that underlie human life (cells and genes) and thereby provide a crucial foundation for biological research and medical application. As with the Human Genome Project, we will also need corresponding atlases for important model organisms, where conserved cell states can be identified and genetic manipulations and other approaches can be used to probe function and lineage. Yet, the Human Cell Atlas differs in important ways from the Human Genome Project: the nature of cell biology means that it will require a distinct experimental toolbox, and will involve making choices concerning molecular and cellular descriptors. Assessing the distance to completion will also be a challenge.

As a Borgesian thought experiment, we could conceive of an imaginary Ultimate Human Cell Atlas that represents: all markers in every cell in a person's body; every cell's spatial position (by adding three dimensions for the body axes); every cell at every moment of a person's lifetime (by adding another dimension for time relating the cells by a lineage); and the superimposition of such cell atlases from every human being, annotated according to differences in health, genotype, lifestyle and environmental exposure.

Of course, it is not possible to construct such an Ultimate Atlas. However, it is increasingly feasible to sample richly from the distribution of points to understand the key features and relationships among all human cells. We return below to the question of how the scientific community might go about creating a Human Cell Atlas. First, we consider the central scientific question: What could we hope to learn from such an atlas?

A Human Cell Atlas would have a profound impact on biology and medicine by bringing our understanding of anatomy, development, physiology, pathology, intracellular regulation, and intercellular communication to a new level of resolution. It would also provide invaluable markers, signatures and tools for basic research (facilitating detection, purification and genetic manipulation of every cell type) and clinical applications (including diagnosis, prognosis and monitoring response to therapy).

In the following sections, we outline reasonable expectations and describe some early examples. We recognize that these concepts will evolve based on emerging data. It is clear that a Human Cell Atlas Project will require and will motivate the development of new technologies. It will also necessitate the creation of new mathematical frameworks and computational approaches that may have applications far beyond biology – perhaps analogous to how biological 'big data' in agriculture in the 1920s led to the creation, by R.A. Fisher and others, of key statistical methods, including the analysis of variance and experimental design (*Parolini, 2015*).

## Taxonomy: cell types

The most fundamental level of analysis is the identification of cell types. In an atlas where cells are represented as points in a high-dimensional space, 'similar' cells should be 'close' in some appropriate sense, although not identical, owing to differences in physiological states (*e.g.*, cell-cycle stage), the inherent noise in molecular systems (*Eldar and Elowitz, 2010*; *Kharchenko et al., 2014*; *Kim et al., 2015*; *Shalek et al., 2013*), and measurement errors (*Buettner et al., 2015*; *Kharchenko et al., 2014*; *Kim et al., 2015*; *Shalek et al., 2013*; *Shalek et al., 2014*; *Wagner et al., 2016*). Thus, a cell 'type' might be defined as a region or a probability distribution (*Kim and Eberwine, 2010*; *Sul et al., 2012*) either in the full-dimensional space or in a projection onto a lower-dimensional space that reflects salient features.

While this notion is intuitively compelling, it is challenging to give a precise definition of a 'cell type'. Cell-type taxonomies are often represented as hierarchies based on morphological, physiological, and molecular differences (*Sanes and Masland, 2015*). Whereas higher distinctions are easily agreed upon, finer ones may be less obvious and may not obey a strict hierarchy, either because distinct types share features, or because some distinctions are graded and not discrete. Critically, it remains unclear whether distinctions based on morphological, molecular and physiological properties agree with each other. New computational methods will be required both to discover types and to better classify cells and, ultimately, to refine the concepts themselves (*Grün and van Oudenaarden, 2015*; *Shapiro et al., 2013*; *Stegle et al., 2015*; *Tanay and Regev, 2017*; *Wagner et al., 2016*). Unsupervised clustering algorithms for high-dimensional data provide an initial framework (*Grün et al., 2015*; *Grün et al., 2016*; *Jaitin et al., 2014*; *Levine et al., 2015*; *Macosko et al., 2015*; *Shekhar et al., 2016*; *Vallejos et al., 2015*), but substantial advances will be needed in order to select the 'right' features, the 'right' similarity metric, and the 'right' level of granularity for the question at hand,

control for distinct biological processes, handle technical noise, and connect novel clusters with legacy knowledge. Whereas cell types are initially defined based on regions in feature space, it will be important eventually to distill them into simpler molecular and morphological signatures that can be used to index cells in the atlas, aggregate and compare results from independent labs and different individuals, and create tools and reagents for validation and follow up studies.

For all the reasons above, we have not attempted to propose a precise definition of 'cell type'. Rather, the definition should evolve based on empirical observation.

Despite these challenges, recent studies in diverse organs – including immune, nervous, and epithelial tissues – support the prospects for comprehensive discovery of cell types, as well as harmonization of genomic, morphological, and functional classifications (*Figure 2A–C*). For example, analysis of immune cells from mouse spleen (*Jaitin et al., 2014*) and human blood (*Horowitz et al., 2013*) showed that well-established functional immune cell types and subtypes could be readily distinguished by unsupervised clustering of single-cell expression profiles. Similarly, single-cell expression profiles of epithelial cells from gut organoids (*Grün et al., 2015*) distinguished known cell subtypes, each with distinctive functional and histological characteristics, while also revealing a new subtype of enteroendocrine cells, which was subsequently validated experimentally.

The nervous system, where many cell types have not yet been characterized by any means, illustrates both the promise and the challenge. Whereas each of the 302 individual neurons in *C. elegans* can be distinctly defined by its lineage, position, connectivity, molecular profile and functions, the extent to which the $\sim 10^{11}$ neurons in the human brain are distinctly defined by morphological, physiological, lineage, connectivity, and electrical-activity criteria, and have distinct molecular profiles, remains unknown. Cellular neuroanatomy is deeply rooted in the concept of cell types defined by their morphologies (a proxy for connectivity) and electrophysiological properties (*Ascoli et al., 2008*), and extensive efforts continue to classify the types in complicated structures like the retina and neocortex (*Jiang et al., 2015*; *Markram et al., 2015*; *Sanes and Masland, 2015*). Critically, it remains unclear whether distinctions based on morphological, connectional, and physiological properties agree with their molecular properties.

The mouse retina provides an ideal testing ground to test this correspondence because cell types follow highly stereotyped spatial patterns (*Macosko et al., 2015*; *Sanes and Masland, 2015*). Analysis of 31,000 retinal bipolar cells (*Shekhar et al., 2016*) automatically re-discovered the 13 subtypes that had been defined over the past quarter-century based on morphology and lamination, while also revealing two new subtypes with distinct morphological and laminar characteristics. These subtypes included one with a 'bipolar' expression pattern and developmental history, but a unipolar morphology in the adult (*Shekhar et al., 2016*), which has distinct functional characteristics in the neural circuits of the retina (*Della Santina et al., 2016*). In this example, known morphological and other non-molecular classifications matched perfectly to molecular types, and new molecularly-defined cell types discovered in the single-cell transcriptomic analysis corresponded to unique new morphology and histology. In other complex brain regions such as the neocortex and hippocampus there are also a large number of transcriptionally defined types (*Darmanis et al., 2015*; *Gokce et al., 2016*; *Habib et al., 2016a*; *Lake et al., 2016*; *Pollen et al., 2014*; *Tasic et al., 2016*; *Zeisel et al., 2015*), but it has been more difficult to find consensus between data modalities, and the relationship between transcriptomic types and anatomical or morphological types is unclear. In this light, technologies that can directly measure multiple cellular phenotypes are essential. For example, electrophysiological measurements with patch clamping followed by scRNA-seq used in a recent study of a particular inhibitory cortical cell type showed that the transcriptome correlated strongly with the cell's physiological state (*Cadwell et al., 2016*; *Földy et al., 2016*). Thus, the transcriptome appears to provide a proxy for other neuronal properties, but much more investigation is needed.

## Histology: cell neighborhood and position

Histology examines the spatial position of cells and molecules within tissues. Over the past century, we have learnt a great deal about cell types, markers, and tissue architecture, and this body of knowledge will need to be further refined and woven seamlessly into the Human Cell Atlas. With emerging highly multiplexed methods for *in situ* hybridization (*Chen et al.,*

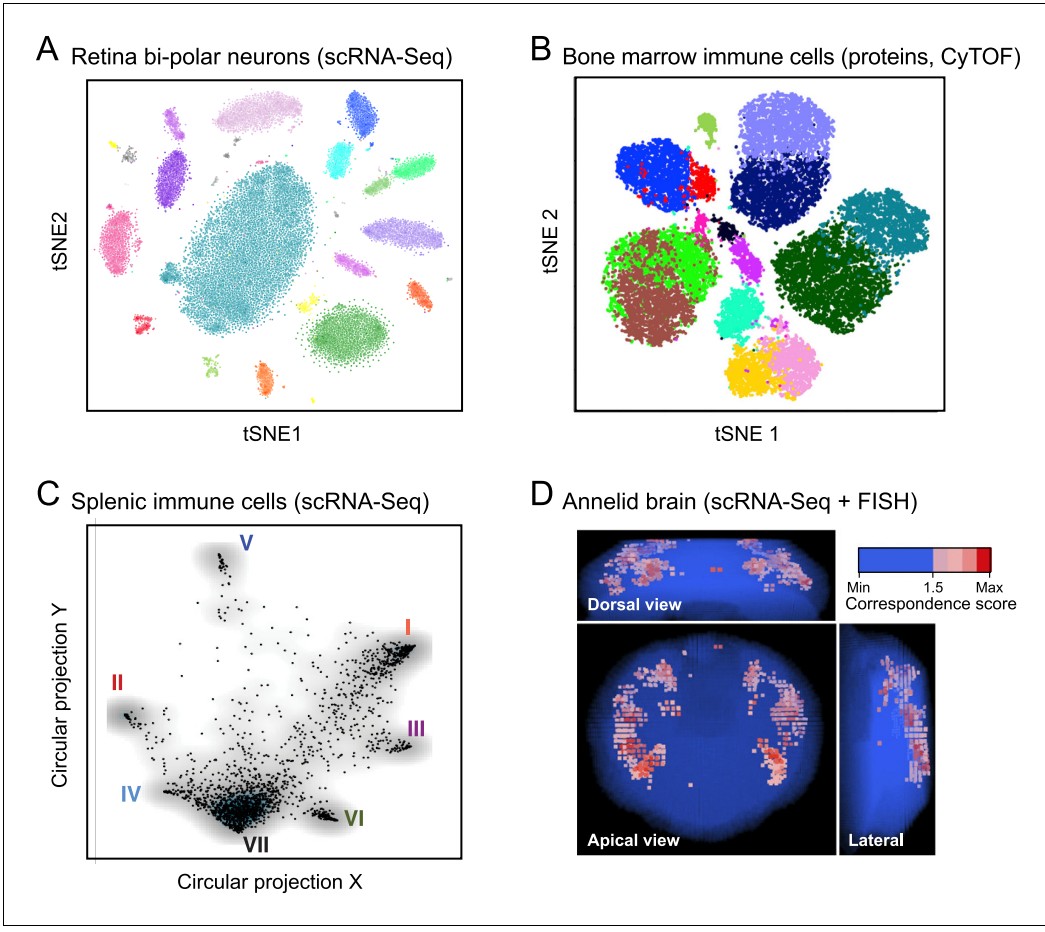

**Figure 2.** Anatomy: cell types and tissue structure. The first three plots show single cells (dots) embedded in low-dimensional space based on similarities between their RNA-expression profiles (**A**, **C**) or protein-expression profiles (**B**), using either t-stochastic neighborhood embedding (**A,B**) or circular projection (**C**) for dimensionality reduction and embedding. (**A**) Bi-polar neurons from the mouse retina. (**B**) Human bone marrow immune cells. (**C**) Immune cells from the mouse spleen. (**D**) Histology. Projection of single-cell data onto tissue structures: image shows the mapping of individual cells onto locations in the marine annelid brain, based on the correspondence (color bar) between their single-cell expression profiles and independent FISH assays for a set of landmark transcripts.

DOI: https://doi.org/10.7554/eLife.27041.005

*2015b*; *Shah et al., 2016*) or protein staining (*Angelo et al., 2014*; *Giesen et al., 2014*), it should be possible to spatially map multiple cell types at once based on expression signatures to see how they relate to each other and to connect them with cell types defined by morphology or function. It should also be possible to extend observations of continuous gradients for individual genes (such as morphogens) to multigene signatures.

Computational approaches could then allow iterative refinement of cellular characterization based on both a cell's molecular profile and information about its neighborhood; methods perfected in the analysis of networks could provide a helpful starting point (*Blondel et al., 2008*; *Rosvall and Bergstrom, 2008*). Conversely, expression data from a cell can help map its position in absolute coordinates or relative terms, as well as in the context of pathology, highlighting how disease tissue differs from typical healthy tissue. Combining molecular profiles with tissue architecture will require new computational methods, drawing perhaps on

advances in machine vision (*Xu et al., 2015*; *Zheng et al., 2015*).

New methods for integrating single-cell genomics data into a spatial context have been developed recently. Single-cell analyses of tissues from early embryos (*Satija et al., 2015*; *Scialdone et al., 2016*) to adult (*Achim et al., 2015*) demonstrate how physical locations can be imprinted in transcriptional profiles (*Durruthy-Durruthy et al., 2014*) and can be used to infer tissue organization (*Figure 2D*). In the early zebrafish embryo, for example, a cell's expression profile specifies its location to within a small neighborhood of ~100 cells; the related expression patterns of individual genes in turn fall into only nine spatial archetypes (*Satija et al., 2015*). In the early mouse embryo, key spatial gradients can be recovered by a 'pseudospace' inferred from reduced dimensions of single cell profiles (*Scialdone et al., 2016*). In adult mouse hippocampus, cell profiles show clear clusters corresponding to discrete functional regions as well as gradients following dorsal/ventral and medial/lateral axes (*Habib et al., 2016a*). In the annelid brain, even finer punctate spatial patterns can be resolved (*Achim et al., 2015*).

## Development: transitions to differentiated cell types

Cells arrive at their final differentiated cell types through partly asynchronous branching pathways of development (*Ferrell, 2012*), which are driven by and reflected in molecular changes, especially gene-expression patterns (see, for example, *Chao et al., 2008*; *Jojic et al., 2013*). It should therefore be possible to reconstruct development as trajectories in high-dimensional space, mirroring Waddington's landscape (*Ferrell, 2012*; *Waddington, 1957*) – just as it would be possible to infer the ski lifts and trails on a mountain from snapshots of the positions of enough skiers. One can even infer sharp transitions, provided enough cells are observed. The required sampling density will depend on the number and complexity of paths and intersections, and sorting strategies can help to iteratively enrich for rare, transient populations. Notably, the relative proportions of cells observed at different points along the developmental paths can help convey critical information, both about the duration of each phase (*Antebi et al., 2013*; *Kafri et al., 2013*) and the balance of how progenitor cells are allocated among fates (*Antebi et al., 2013*; *Lönnberg et al., 2017*; *Moris et al., 2016*),

especially when information about the rate of cell proliferation and/or death can be incorporated as inferred from the profiles.

In animal models, it should be possible to create true lineage trees by marking a common progenitor cell type. For example, one might use synthetic circuits that introduce a molecular barcode only in cells expressing an RNA pattern characteristic of the cell type in order to recognize its descendants (*Gagliani et al., 2015*; *McKenna et al., 2016*). In humans, immune cells naturally contain lineage barcodes through VDJ recombination in T and B cells and somatic hypermutation in B cells (*Stubbington et al., 2016*). More generally, it may be feasible to accomplish lineage tracing in human cells by taking advantage of the steady accumulation of DNA changes (such as somatic point mutations, or repeat expansions at microsatellite loci) at each cell division (*Behjati et al., 2014*; *Biezuner et al., 2016*; *Martincorena et al., 2015*; *Reizel et al., 2012*; *Shlush et al., 2012*) or as a molecular clock (*Taylor et al., 2003*; *Teixeira et al., 2013*).

Initial computational methods have already been developed for inferring dynamic trajectories from large numbers of single-cell profiles, although better algorithms are still needed. Critical challenges include accurately inferring branching structures, where two or more paths diverge from a single point; reconstructing 'fast' transitions, where only few cells can be captured; and accounting for the fact that a cell may be following multiple dynamic paths simultaneously – for example, differentiation, the cell cycle, and pathogen response (see below) – that may affect each other. The reconstruction algorithms themselves could incorporate insights from theoretical studies of dynamical systems, and learned models could be analyzed in light of such frameworks (*Ferrell, 2012*; *May, 1976*; *Thom, 1989*).

Recent studies provide proofs-of-principle for how simultaneous and orthogonal biological processes can be inferred from single-cell RNA-seq data (*Figure 3*; *Angerer et al., 2016*; *Bendall et al., 2014*; *Chen et al., 2016b*; *Haghverdi et al., 2015*; *Haghverdi et al., 2016*; *Lönnberg et al., 2017*; *Marco et al., 2014*; *Moignard et al., 2015*; *Setty et al., 2016*; *Trapnell et al., 2014*; *Treutlein et al., 2016*). Linear developmental trajectories have been reconstructed, for example, from single-cell protein expression during B-cell differentiation (*Bendall et al., 2014*), and from single-cell RNA expression during myogenesis *in vitro*

(*Trapnell et al., 2014*), early hematopoiesis (*Nestorowa et al., 2016*), neurogenesis *in vivo* (*Habib et al., 2016a*; *Shin et al., 2015*), and reprogramming from fibroblasts to neurons (*Treutlein et al., 2016*). With a large enough number of cells, analysis of B-cell development was able to highlight a rare (0.007%) population corresponding to the earliest B-cell lymphocytes and confirm the identification by reference to

rearrangements at the IgH locus. In direct reprogramming to neurons, scRNA-seq revealed unexpected trajectories (*Treutlein et al., 2016*). Bifurcated trajectories have also been reconstructed in the differentiation of embryonic stem cells, T helper cells, and hematopoietic cells (*Chen et al., 2016b*; *Haghverdi et al., 2015*; *Haghverdi et al., 2016*; *Lönnberg et al., 2017*; *Marco et al., 2014*; *Moignard et al., 2015*;

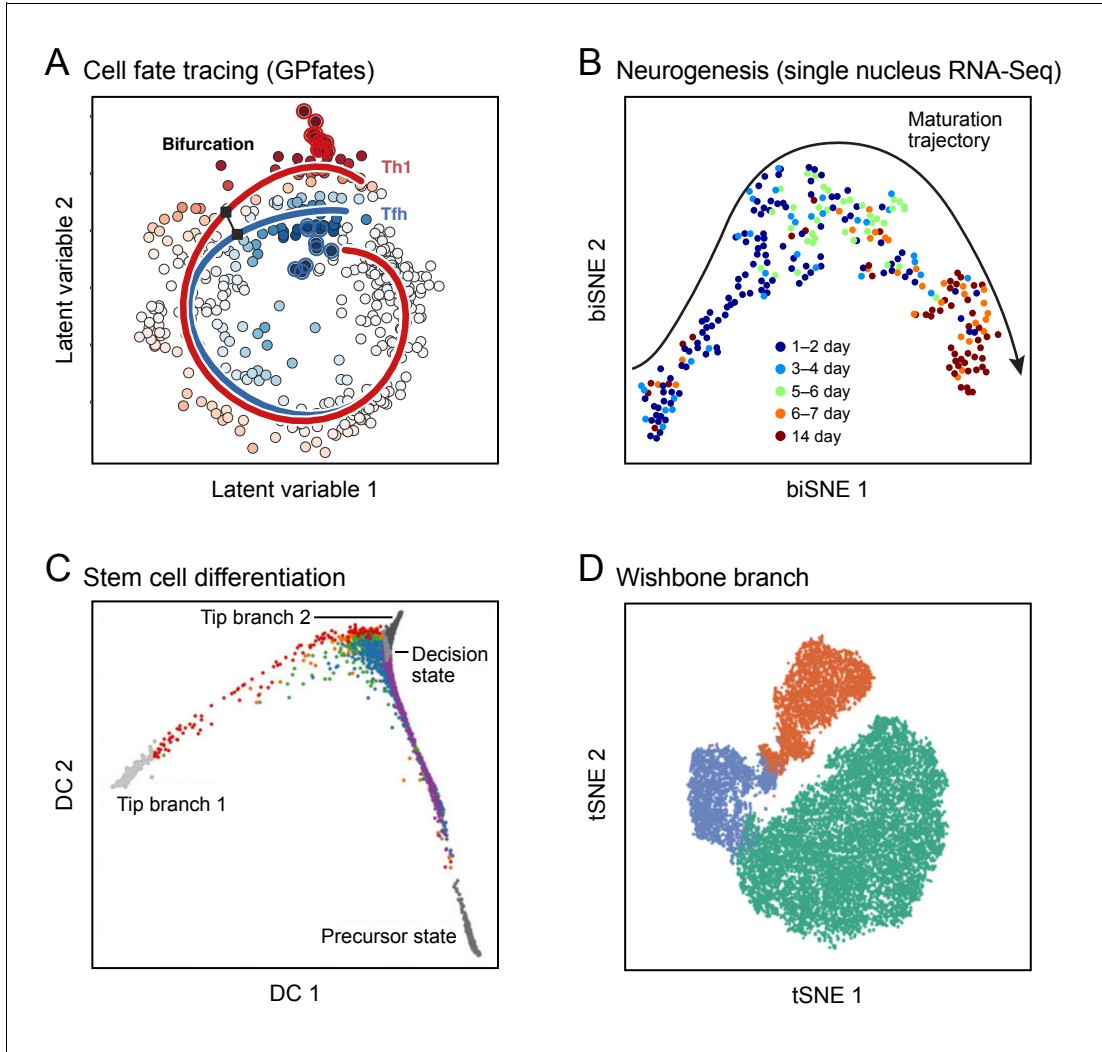

**Figure 3.** Developmental trajectories. Each plot shows single cells (dots; colored by trajectory assignment, sampled time point, or developmental stage) embedded in low-dimensional space based on their RNA (**A-C**) or protein (**D**) profiles, using different methods for dimensionality reduction and embedding: Gaussian process patent variable model (**A**); t-stochastic neighborhood embedding (**B, D**); diffusion maps (**C**). Computational methods then identify trajectories of pseudo-temporal progression in each case. (**A**) Myoblast differentiation in vitro. (**B**) Neurogenesis in the mouse brain dentate gyrus. (**C**) Embryonic stem cell differentiation in vitro. (**D**) Early hematopoiesis.

DOI: https://doi.org/10.7554/eLife.27041.006

*Setty et al., 2016*), and have helped address open questions about whether myeloid progenitor cells in bone marrow are already skewed towards distinct fates (*Olsson et al., 2016*; *Paul et al., 2015*) and when T helper cell commit to their fate (*Lönnberg et al., 2017*).

## Physiology and homeostasis: cycles, transient responses and plastic states

In addition to development and differentiation, cells are constantly undergoing multiple dynamic processes of physiological change and homeostatic regulation (*Yosef and Regev, 2011*; *2016*). These include *cyclical processes*, such as the cell cycle and circadian rhythms; *transient responses* to diverse factors, from nutrients and microbes to mechanical forces and tissue damage; and *plastic states* that can be stably maintained over longer time scales, but can change in response to new environmental cues. (The precise boundary between plastic states and cell types, it must be noted, remains to be clarified.) The molecular phenotype of a cell reflects a superposition of these various processes and their interactions (*Wagner et al., 2016*).

Studies of physiological processes from bulk tissue samples are hampered by asynchrony and heterogeneity among cells, which blur the signals of individual processes and states; investigators strive to create homogeneous cell populations through synchronization and purification. By contrast, single-cell analysis exploits asynchrony and heterogeneity, leveraging variation within a cell population to reveal underlying structures. The difference is analogous to two approaches in structural biology: X-ray crystallography, which requires molecules to be in a crystalline order, and cryo-electron microscopy, which depends on observing large numbers of molecules in randomly sampled poses.

From asynchronous observations of cyclical and transient processes, it should be possible to 'order' cells with respect to the process (as for development), with cell proportions reflecting residence time (*e.g.*, the length of a phase of the cell cycle). As was initially shown for single-cell measurement of a few features of the cell cycle (*Kafri et al., 2013*), analysis of many systems could yield a near-continuous model of the process, provided that a sufficient number of cells is sampled. This can occur either because all phases co-occur (*e.g.*, in asynchronously cycling cells) or because enough time points are sampled to span the full process. If very rapid

and dramatic discontinuities exist, recovering them would likely require direct tracing, for example by genetic tracers or live analysis in cell cultures, organoids, or animal models.

Once the cells are ordered, one can derive gene-signatures that reflect each phase and use them to further sharpen and refine the model. With sufficient data, it should also be possible to tease apart interactions among processes occurring in parallel (such as the cell cycle, response to a pathogen, and differentiation). For plastic states, it may be possible to capture transient transitions between them, especially if they can be enriched by appropriate physiological cues. Finally, we will likely learn about the nature of stable states: while we often think of stable states as discrete attractor basins (*Waddington, 1957*), there may also be troughs that reflect a continuous spectrum of stable states (*e. g.*, the ratio of two processes may vary across cells, but are stable in each; *Antebi et al., 2013*; *Gaublomme et al., 2015*; *Huang, 2012*, *2013*; *Rebhahn et al., 2014*; *Zhou et al., 2012*; *Zhou et al., 2016*). Some key aspects of processes may be difficult to uncover solely from observations of transitions among molecular states, and will likely require directed perturbations and detailed mechanistic studies.

Recent studies have shown that cyclical processes and transient responses – from the cell cycle (*Buettner et al., 2015*; *Gut et al., 2015*; *Kafri et al., 2013*; *Kowalczyk et al., 2015*; *Macosko et al., 2015*; *Proserpio et al., 2016*; *Tirosh et al., 2016a*) to the response of immune cells to pathogen components (*Avraham et al., 2015*; *Shalek et al., 2013*; *Shalek et al., 2014*) – can be traced in single-cell profiles. It is possible to order the cells temporally, define coordinately expressed genes with high precision, identify the time scale of distinct phases, and relate these findings to orthogonal measures (*Figure 4*). For example, in the cell cycle, analysis of single-cell profiles readily shows a robust, reproducible and evolutionarily conserved program that can be resolved in a near-continuous way across human and mouse cell lines (*Macosko et al., 2015*), primary immune cells (*Buettner et al., 2015*; *Kowalczyk et al., 2015*), and healthy and disease tissues (*Patel et al., 2014*; *Tirosh et al., 2016a*; *Tirosh et al., 2016b*). This approach has made it possible to determine the relative rates of proliferation of different cell subpopulations within a dataset (*Buettner et al., 2015*; *Kolodziejczyk et al., 2015*; *Kowalczyk et al., 2015*; *Tsang et al., 2015*), a feat difficult to accomplish using bulk

synchronized populations along the cell cycle (*Bar-Joseph et al., 2008*; *Lu et al., 2007*). Notably, the cell cycle could also be reconstructed by similar approaches when applied to imaging data of very few molecular markers along with salient spatial features (*Gut et al., 2015*) or with morphology alone (*Blasi et al., 2016*; *Eulenberg et al., 2017*). Similar principles apply to transient responses. In the response of dendritic cells to pathogen components, single-cell profiling uncovered a small subset (<1%) of 'precocious' cells: these early-appearing cells express a distinctive module of genes, initiate production of interferon beta, and coordinate the subsequent response of other cells through paracrine signaling (*Shalek et al., 2014*).

## Disease: cells and cellular ecosystems

The Human Cell Atlas will be a critical reference for studying disease, which invariably involves disruption of normal cellular functions, interactions, proportions, or ecosystems. The power of single-cell analysis of disease is evident from decades of histopathological studies and FACS analysis. It will be substantially extended by the routine ability to characterize cells and tissues with rich molecular signatures, rather than focusing on a limited number of pre-defined markers or cell populations. It will also support the growing interest in understanding interactions

between frankly abnormal cells and all other cells in a tissue's ecosystem in promoting or suppressing disease processes (*e.g.*, between malignant cells and the tumor microenvironment).

Single-cell analysis of disease samples will also likely be critical to see the full range of normal cellular physiology, because disease either elicits key perturbs cellular circuitry in informative ways. A clear example is the immune system, where only in the presence of a 'challenge' is the full range of appropriate physiological behaviors and potential responses by a cell revealed.

Single-cell information across many patients will allow us to learn about how cell proportions and states vary and how this variation correlates with genome variants, disease course and treatment response. From initial studies of a limited number of patients, it should be possible to derive signatures of key cell types and states and use them to deconvolute cellular proportions in conventional bulk-tissue or blood samples (*Levine et al., 2015*; *Tirosh et al., 2016a*). Future studies may expand single-cell analysis to thousands of patients to directly investigate how genetic variation affects gene transcription and regulation.

The hematopoietic system will be an early and fruitful target. A study involving signatures of cell-signaling assays by single-cell mass cytometry of healthy hematopoietic cells led to

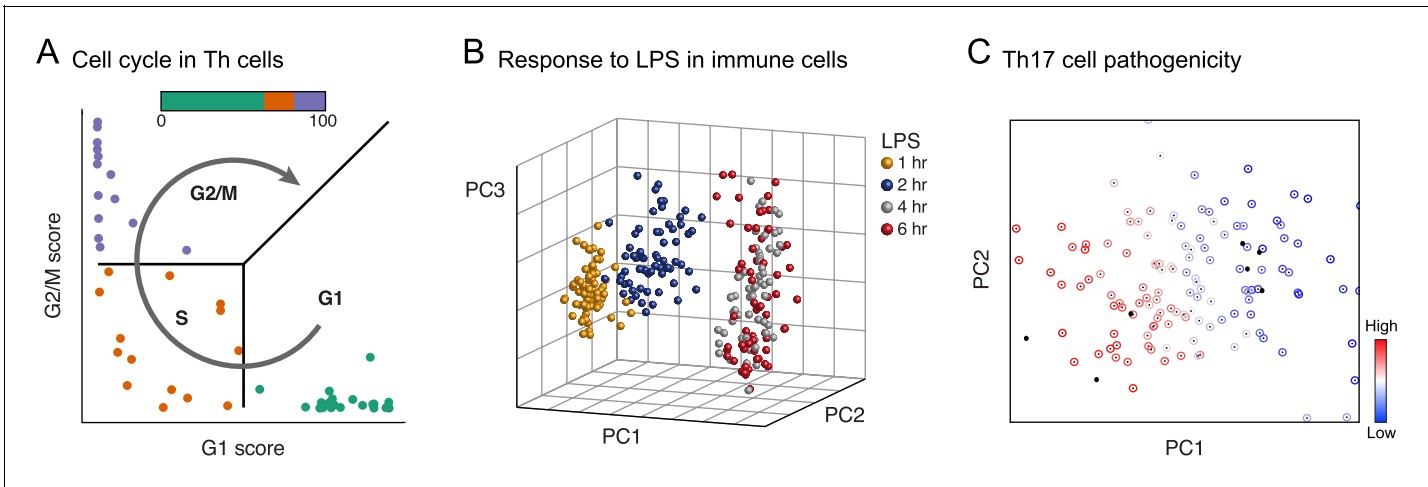

**Figure 4.** Physiology. Each plot shows single cells (dots) embedded in low-dimensional space on the basis of their RNA profile, based on predefined gene signatures (**A**) or PCA (**B**, **C**), highlighting distinct dynamic processes. (**A**) The cell cycle in mouse hematopoietic stem and progenitor cells; adapted under terms of CC BY 4.0 from *Scialdone et al. (2015)*. (**B**) Response to lipopolysaccharide (LPS) in mouse immune dendritic cells. (**C**) Variation in the extent of pathogenicity in mouse Th17 cells.
DOI: https://doi.org/10.7554/eLife.27041.007

more accurate classification of hematopoietic stem and progenitor cells (HSPCs) in Acute Myeloid Leukemia; a previous classification was error-prone, because the 'classical' cell-surface markers of healthy cells do not correctly identify the corresponding population in disease, whereas a richer signature allows accurate identification (*Levine et al., 2015*). Monitoring rare immune populations first discovered in a normal setting can help zero in on the relevant aberrations in disease. For example, the rare population associated with VDJ recombination first identified by trajectory analysis of B cell development (*Bendall et al., 2014*) is expanded in pediatric Acute Lymphoblastic Leukemia, and drastically more so in recurrence (Gary Nolan, unpublished results).

The greatest impact, at least in the short term, is likely to be in cancer. Early studies used single-cell qPCR to investigate the origin of radioresistance in cancer stem cells (*Diehn et al., 2009*) and to dissect the heterogeneity and distortions of cellular hierarchy in colon cancer (*Dalerba et al., 2011*). With the advent of high-throughput methods, single-cell genome analysis has been used to study the clonal structure and evolution of tumors in both breast cancer (*Wang et al., 2014*) and acute lymphoblastic leukemia (*Gawad et al., 2014*), and to infer the order of earliest mutations that cause acute myeloid leukemia (*Corces-Zimmerman et al., 2014*; *Jan et al., 2012*).

In recent studies of melanoma (*Tirosh et al., 2016a*), glioblastoma (*Patel et al., 2014*), low-grade glioma (*Tirosh et al., 2016b*), and myeloproliferative neoplasms (*Kiselev et al., 2017*), single-cell RNA-seq of fresh tumors resected directly from patients readily distinguished among malignant, immune, stromal and endothelial cells. Among the malignant cells, it identified distinct cell states – such as cancer stem cells (*Patel et al., 2014*; *Tirosh et al., 2016b*), drug-resistant states (*Tirosh et al., 2016a*), proliferating and quiescent cells (*Patel et al., 2014*; *Tirosh et al., 2016a*; *Tirosh et al., 2016b*) – and related them to each other, showing, for example, that only stem-like cells proliferate in low-grade glioma (*Tirosh et al., 2016b*) and that individual sub-clones can be readily identified in one patient (*Kiselev et al., 2017*). Among the non-malignant cells, it found distinct functional states for T-cells, and revealed that, while activation and exhaustion programs are coupled, the exhausted state is also controlled by an independent regulatory program in both human tumors (*Tirosh et al., 2016a*) and a mouse model

(*Singer et al., 2016*). To associate patterns observed in a few (5-20) patients with effects on clinical phenotypes, single-cell based signatures were used to deconvolute hundreds of bulk tumor profiles that had been collected with rich clinical information (*Levine et al., 2015*; *Patel et al., 2014*; *Tirosh et al., 2016a*).

## Molecular mechanisms: intracellular and inter-cellular circuits

A Human Cell Atlas can also shed light on the molecular mechanisms that control cell type, differentiation, responses and states – within cells, between cells, as well as between cells and their tissue matrix.

For example, over the past several decades, biologists have sought to infer the circuitry underlying gene regulation by observing correlations between the expression of particular regulators and specific cellular phenotypes, drawing inferences about regulation, and testing their models through targeted genetic perturbations. Single-cell data provide a massive increase not only in the quantity of observations, but also in the range of perturbations. The number of cells profiled in a single-cell RNA-seq experiment can far exceed the number of profiles produced even by large consortia (such as ENCODE, FANTOM, TCGA, and GTEx). Moreover, each single cell is a perturbation system in which the levels of regulatory molecules vary naturally – sometimes subtly, sometimes dramatically – due to both stochastic and controlled phenomena within a single genetic background, providing rich information from which to reconstruct cellular circuits (*Krishnaswamy et al., 2014*; *Sachs et al., 2005*; *Shalek et al., 2013*; *Stewart-Ornstein et al., 2012*).

Initial studies have shown that such analyses can uncover intracellular regulators governing cell differentiation and response to stimuli. For example, co-variation of RNA levels across a modest number of cells from a relatively 'pure' population of immune dendritic cells responding to a pathogen component was sufficient to connect antiviral transcription factors to their target genes, because of asynchrony in the responses (*Shalek et al., 2013*). Similarly, co-variation analysis of a few hundred Th17 cells spanning a continuum from less to more pathogenic states revealed regulators that control pathogenicity, but not other features, such as cell differentiation (*Gaublomme et al., 2015*). Co-variation identified a role for pregnenolone biosynthesis

in the response of Th2 cells to helminth infection (*Mahata et al., 2014*), and new regulators of pluripotency in mouse embryonic stem cells (*Kolodziejczyk et al., 2015*). Computationally ordering cells along a time-course of development provides another way to infer regulators – a strategy that has been successful in, for example, differentiating B cells (*Bendall et al., 2014*), myoblasts (*Trapnell et al., 2014*), neurons (*Habib et al., 2016a*; *Shin et al., 2015*), and T helper cells (*Lönnberg et al., 2017*). Finally, when circuitry is already known, variation across single cells can be used to infer exquisite – and functionally important – quantitative distinctions about how signal is processed and propagated. An elegant example is a recent analysis of signaling pathways downstream from the T cell receptor, where single-cell proteomics data has shown how the same cellular circuitry processes signals differently in naïve and antigen-exposed T cells (*Krishnaswamy et al., 2014*).

Beyond transcriptome analysis, single-cell multi-omic profiles (*Box 1*) will improve the inference of cellular circuitry by connecting regulatory mechanisms and their targets (*Tanay and Regev, 2017*). For example, simultaneous measurement of chromatin accessibility and RNA levels may help identify which regulatory regions – and by inference which *trans*–acting regulators – control the levels of which genes. Concomitant measurement of DNA mutations and transcriptional profiles in cancer cells may allow similar causal connections to be drawn, as has been recently shown for mutations in the *CIC* gene and the expression of its regulatory targets (*Tirosh et al., 2016b*).

Studies can be extended from naturally occurring variation among cells to engineered perturbations, by using pooled CRISPR libraries to manipulate genes and reading out both the perturbation and its effects on cellular phenotype in single cells – for example, by single-cell RNA-Seq (*Adamson et al., 2016*; *Dixit et al., 2016*; *Jaitin et al., 2016*).

A cell atlas can also help shed light on intercellular communication, based on correlated profiles across cell types and patients. For example, analysis of single-cell profiles from many small clusters of a few aggregated cells allowed the construction of a cell-cell interaction network in the bone marrow, uncovering specific interaction between megakaryocytes and neutrophils, as well as between plasma cells and neutrophil precursors (Alexander van Oudenaarden, unpublished results). Cell-cell interactomes have also been inferred from profiles of purified cell populations, based on the secreted and cell surface molecules that they express (*Ramilowski et al., 2015*).

In tumors from melanoma patients, gene-expression analysis (involving single-cell data obtained from some patients and bulk tumor data from many more patients, deconvoluted based on signatures learned from the single cells) found genes that are expressed in one cell type, but whose expression levels are correlated with the proportion of a different cell type that does not express them; this analysis revealed that high expression of the complement system in cancer-associated fibroblasts in the tumor microenvironment is correlated with increased infiltration of T cells (*Tirosh et al., 2016a*). Analysis of individual subcutaneous adipose stem cells revealed the existence of a novel cell population that negatively controls the differentiation of the resident stem cells into adipocytes, thus influencing adipose tissue growth and homeostasis (Bart Deplancke, unpublished results). In breast cancer tissues, spatial analysis of multiplex protein expression by imaging mass cytometry (*Giesen et al., 2014*) allowed classification of infiltrating immune cells and malignant cells based on the neighborhood of surrounding cells, highlighting new functional interactions (Bernd Bodenmiller, personal communication).

## A user's guide to the Human Cell Atlas: applications in research and medicine

The Human Genome Project had a major impact on biomedicine by providing a comprehensive reference, a DNA sequence in which answers could be readily looked up and from which unique 'signatures' could be derived (*e.g.*, to recognize genes on microarrays or protein fragments in mass spectrometry). A Human Cell Atlas could provide similar benefits from basic research to clinically relevant applications.

Scientists will be able, for example, to look up precisely in which cell types a gene of interest is expressed and at which level. Today, it is surprisingly challenging to obtain definitive answers for most human genes beyond tissue- or organ-level resolution, although there have been pioneering efforts for the brain and immune system in mouse (*Bakken et al., 2016*; *Hawrylycz et al., 2012*; *Kim and Lanier, 2013*; *Miller et al., 2014*) and for protein expression in human (*Thul et al., 2017*; *Uhlén et al., 2015*). Yet, the question is of enormous importance to basic biologists studying development or

comparing a model system to human biology, medical scientists examining the effect of a disease-causing mutation, and drug developers concerned about the potential toxicities of a small molecule or a CAR-T cell targeting a specific protein (*Brudno and Kochenderfer, 2016*).

Researchers will also be able to derive expression signatures that uniquely identify cell types. Such signatures provide a starting point for a vast range of experimental assays – from molecular markers for isolating, tagging, tracing or manipulating cells in animal models or human samples, to characterization of the effect of drugs on the physiological state of a tissue. Such descriptors of cellular identity will be widely used in clinical assays. For example, today's Complete Blood Count (CBC), a census of a limited number of blood components, may be supplemented by a 'CBC 2.0' that provides a high-resolution picture of the nucleated cells, including the number and activity states of each type in comparison with healthy reference samples. Analogous measures should be possible for other tissues as well. For example, gut biopsies from patients with ulcerative colitis or colon cancer could be analyzed for the type, response, state and location of each of the diverse epithelial, immune, stromal and neural cells that comprise them.

## Toward a Human Cell Atlas

How might the biomedical community build a Human Cell Atlas? As with the Human Genome Project, a robust plan will best emerge from wide-ranging scientific discussions and careful planning involving biologists, technologists, pathologists, physicians, surgeons, computational scientists, statisticians, and others. As noted above, various discussions have taken place for over two years about the idea of a comprehensive Human Cell Atlas, as well as about specific atlases for the brain and the immune system. Several pilot efforts are already underway. Moreover, over the past year discussions have been underway to create an initial plan for a Human Cell Atlas Project (which is articulated in the White Paper mentioned above). Among the key points for consideration are the following:

### Phasing of goals

While the overall goal is to build a comprehensive atlas with diverse molecular measurements, spatial organization, and interpretation of cell types, histology, development, physiology and molecular mechanisms, it will be wise to set intermediate goals for 'draft' atlases at increasing resolution, comprehensiveness, and depth of interpretation. The value of a phased approach was illustrated by the Human Genome Project, which defined milestones along the way (genetic maps, physical maps, rough-draft sequence, finished sequence) that held the project accountable and provided immediate utility to the scientific community.

### Sampling strategies

While an adult human has ~2 x $10^{13}$ nucleated cells, it is neither possible nor necessary to study them all to recover the fine distinctions among human cells. The key will be to combine sound statistical sampling, biological enrichment purification, and insights from studies of model organisms. It is likely beneficial to apply an adaptive, iterative approach with respect to both the number of cells and depth of profiles, as well as anatomical coverage and spatial resolution in the tissue, with initial sparse sampling driving decisions about further sampling. This adaptive approach, termed a 'Sky Dive', adjusts as resolution increases (and is further described in the HCA White Paper).

Such approaches can be facilitated by experimental techniques that allow fast and inexpensive 'banking' of partially processed samples, to which one can return for deeper analysis as methods mature. Advances in handling fixed or frozen tissues would further facilitate the process (*Box 1*). With respect to depth of profiling, recent studies suggest the utility of a mixed strategy: relatively low coverage of the transcriptome can identify many cell types reliably (*Heimberg et al., 2016*; *Shekhar et al., 2016*) and a smaller set of deep profiles can be help interpret the low-coverage data to further increase detection power. As a result, the 'Sky Dive' begins with large-scale uniform sampling, follows with stratified sampling, and then employs specialized sampling at lower throughput.

### Breadth of profiles

The atlas must combine two branches – a cellular branch, focused on the properties of individual cells, and a spatial branch, describing the histological organization of cells in the tissue. For the cellular branch, massively parallel transcriptome analysis of individual single cells or nuclei will likely be the workhorse for efforts in the first few years. However, other robust, high-throughput

profiling methods are rapidly emerging, including techniques for studying chromatin, genome folding, and somatic mutations at single-cell resolution (*Box 1*). For the spatial branch, *in situ* analysis of the spatial patterns of RNA, proteins, and potentially epigenomics will be equally important. While some of these methods are already rapidly maturing, others will benefit from focused development efforts, as well as from cross-comparison among different techniques. Fortunately, most can be applied to preserved tissue specimens, allowing specimens collected now to be analyzed later, as methods mature.

### Biological scope

It will be important to consider the balance among tissue samples from healthy individuals at various stages; small cohorts of individuals with diseases; and samples from model organisms, where key developmental stages are more accessible and manipulations more feasible. Well-chosen pilot projects could help refine strategies and galvanize communities of biological experts. Some communities and projects would be organized around organs (*e.g.*, liver, heart, brain), others around systems (*e.g.*, the immune system) or disease (*e.g.*, cancer), the latter distributed across many organs and tissues.

As outlined in the HCA White Paper, the first draft of the atlas might pursue roughly a dozen organs and systems, each from up to 100 individuals, collected across 3–4 geographical sites; each would be analyzed to obtain both cellular and spatial data, by means of uniform to stratified sampling. Tissue from post-mortem examination will play a key role, because it is the only way to obtain samples from a single individual across the entire body. These efforts will be complemented, where possible, by biopsy or resection material from healthy research participants, and by whole organs obtained from deceased transplant donors after transplantation organs have been harvested. In some cases, such as the immune system, samples from individuals with a disease will be included to probe different functional states of a system.

The full atlas, will ultimately describe at least 10 billion cells, covering all tissues, organs, and systems. Specimens will come from both healthy research participants and small cohorts of patients with relevant diseases. The cells and tissues will be studied using a broad range of techniques, to obtain cellular and spatial information, from samples designed to represent the world's diversity. As with previous genomic projects, the Human Cell Atlas will be bounded in its resolution (with respect to the rarity of cell types/states and the spatial resolution), its coverage of disease and diversity (broadly representative but not obviating the need for direct genetic and clinical studies), and its functional assessment (to validate the existence of identified cells and facilitate – but not perform – detailed functional characterization).

### Model organisms

The Human Genome Project and the broader scientific community benefitted from insights learned from genome projects conducted in parallel in model organisms. These projects empowered functional studies in model organisms, ushered a new era of comparative genomics, and provided important technical lessons. By analogy, we envision that key 'sister' atlases in model organisms will be developed in parallel and in coordination with the Human Cell Atlas. These projects should not delay progress on the human atlas (or *vice versa*), because current techniques are already directly applicable to biomedical research on human samples.

In some cases, model organism atlases can use techniques that are not possible in humans, such as engineering animals to facilitate lineage tracing. In many cases, the extensive validation and functional follow-up studies that can be performed in model organisms will help validate 'by proxy' conclusions drawn in the human atlas. Finally, comparing the atlases across organisms will provide invaluable lessons in evolution and function.

### Quality

In creating a reference map to be used by thousands of investigators, it is critical to ensure that the results are of high quality and technically reproducible. This is especially important in view of the inherent biological variation and expected measurement noise. Substantial investment will be needed in the development, comparison, and dissemination of rigorous protocols, standards, and benchmarks. Both individual groups and larger centers will likely have important roles in defining and ensuring high quality. It will also be important that the collected samples be accompanied by excellent clinical annotations, captured in consistent meta-data across the atlas.

Tissue processing poses special challenges, including the need for robust methods for dissociating samples into single cells so as to preserve all cell types, fixation for *in situ* methods, and

freezing for transport. A related challenge is the difference in the amenability of specific cell types for different assays (T cells are very small and yield lower quality scRNA-seq; the fat content in adipocyte is challenging for many spatial methods; many neurons cannot currently be isolated with their axons and dendrites from adult tissue). Careful attention will also be needed to data generation and computational analysis, including validated standard operating procedures for experimental methods, best practices, computational pipelines, and benchmarking samples and data sets to ensure comparability.

### Global equity

Geographical atlases of the Earth were largely developed to serve global power centers. The Human Cell Atlas should be designed to serve all people: it should span genders, ethnicities, environments, and the global burden of diseases – all of which are likely to affect the molecular profiles of cells and must be characterized to maximize the atlas's benefits. The project itself should encourage and support the participation of scientists, research centers and countries from around the globe, while recognizing the value of respecting and learning from diverse populations, cultures, mores, beliefs, and traditions.

### Open data

The Human Genome Project made clear the power of open data that can be used by all and freely combined with other datasets. A Human Cell Atlas should similarly be an open endeavor, to the full extent permitted by participants' wishes and legal regulation. While the underlying sequence data contains many polymorphisms that make it 'identifiable', it should be possible to map the data onto 'standard models' of each gene to substantially mitigate this issue. To make the atlas useful, it will be critical to develop data platforms that can provide efficient aggregation and storage, quality control, analytical software, and user-friendly portals.

### Flexibility

A Human Cell Atlas Project should be intellectually and technologically flexible. The project should embrace the fact that its biological goals, experimental methods, computational approaches, overall scale, and criteria for 'completion' will evolve rapidly as insights and tools develop. For historical context, it is useful to remember that discussions about a Human Genome Project began before the development of automated DNA sequencing machines, the polymerase chain reaction, or large-insert DNA cloning, and the project drove technological progress on many fronts. Moreover, the criteria for a 'finished' genome sequence were only agreed upon during the last third of the project.

### Impact on the scientific community

Large-scale efforts, such as a Human Cell Atlas, must be careful to appropriately weigh the benefits to science and individual scientists with the potential costs. We consider the key benefits to the broad scientific community to include: the core scientific knowledge and discoveries that will result from having a reference map; the empowerment of scientists working across any tissue or cell type to pursue their research more precisely and effectively; the development, hardening and dissemination of experimental techniques and computational methods in the context of big-data settings, all of which will be openly shared; the inclusive and maximally open Human Cell Atlas community, inviting participation by all individual labs and research centers; and the coordination of efforts that would otherwise be unconnected, less extensive, and more expensive.

At the same time, we must be aware of potential pitfalls, including: premature restriction to specific technologies or approaches, which might limit innovation in a fast-moving field; implicit restriction of participation, based on available resources; and diversion of funding from other research directions. The unique organization and community of the Human Cell Atlas Project will tackle these potential challenges by committing to open membership, to the open and immediate data release with no restrictions, and to open-source code for all computational approaches. We hope that the new information and technology generated will more than repay the costs of the project by increasing the speed and efficiency of biomedical research throughout the scientific community.

### Engagement with the non-scientific community

The general public is a key stakeholder community for the Human Cell Atlas. Proper public engagement should involve many communities, including interested members of the public, citizen-scientists, schoolchildren, teachers and, where appropriate, research participants. Engagement will take diverse forms, including traditional media, social media, video and,

importantly, direct sharing of the project's data. Across all channels, it will be important to articulate the goals, principles and motivations of the project. While explaining the intended benefits to the public with respect to advancing disease biology, drug discovery and diagnostics, it will be equally important to avoid 'hype': that is, we need to avoid making promises and raising expectations that are unrealistic in content or timing.

### Forward looking

Any data produced today will be easier, faster, more accurate and cheaper to produce tomorrow. Any intermediate milestones achieved during the project will be supplanted by deeper, broader, more accurate, and more comprehensive successors within a few short years. However, as we define the goal of a Human Cell Atlas Project, we should view it not as a final product, but as a critical stepping-stone to a future when the study of human biology and medicine is increasing tractable.

## Conclusion

The past quarter-century has shown again and again the value of the scientific community joining together in collaborative efforts to generate and make freely available systematic information resources to accelerate scientific and medical progress in tens of thousands of laboratories around the world. The Human Cell Atlas builds on this rich tradition, extending it to the fundamental unit of biological organization: the cell.

Many challenges will arise along the way, but we are confident that they can be met through scientific creativity and collaboration. It is time to begin.

**Aviv Regev** Broad Institute of MIT and Harvard, Cambridge, United States; Department of Biology, Massachusetts Institute of Technology, Cambridge, United States; Howard Hughes Medical Institute, Chevy Chase, United States
aregev@broadinstitute.org
http://orcid.org/0000-0003-3293-3158

**Sarah A Teichmann** Wellcome Trust Sanger Institute, Hinxton, United Kingdom; EMBL-European Bioinformatics Institute, Hinxton, United Kingdom; Cavendish Laboratory, Department of Physics, University of Cambridge, Cambridge, United Kingdom
st9@sanger.ac.uk

**Eric S Lander** Broad Institute of MIT and Harvard, Cambridge, United States; Department of Biology, Massachusetts Institute of Technology, Cambridge, United States; Department of Systems Biology, Harvard Medical School, Boston, United States
eric@broadinstitute.org

**Ido Amit** Department of Immunology, Weizmann Institute of Science, Rehovot, Israel

**Christophe Benoist** Division of Immunology, Department of Microbiology and Immunobiology, Harvard Medical School, Boston, United States

**Ewan Birney** EMBL-European Bioinformatics Institute, Wellcome Genome Campus, Hinxton, United Kingdom

**Bernd Bodenmiller** Institute of Molecular Life Sciences, University of Zürich, Zürich, Switzerland

**Peter Campbell** Wellcome Trust Sanger Institute, Wellcome Genome Campus, Hinxton, United Kingdom; Department of Haematology, University of Cambridge, Cambridge, United Kingdom

**Piero Carninci** Division of Genomic Technologies, RIKEN Center for Life Science Technologies, Yokohama, Japan
https://orcid.org/0000-0001-7202-7243

**Menna Clatworthy** Molecular Immunity Unit, Department of Medicine, MRC Laboratory of Molecular Biology, University of Cambridge, Cambridge, United Kingdom

**Hans Clevers** Hubrecht Institute, Princess Maxima Center for Pediatric Oncology and University Medical Center Utrecht, Utrecht, The Netherlands

**Bart Deplancke** Institute of Bioengineering, School of Life Sciences, Swiss Federal Institute of Technology (EPFL), Lausanne, Switzerland
https://orcid.org/0000-0001-9935-843X

**Ian Dunham** EMBL-European Bioinformatics Institute, Wellcome Genome Campus, Hinxton, United Kingdom

**James Eberwine** Department of Systems Pharmacology and Translational Therapeutics, Perelman School of Medicine, University of Pennsylvania, Philadelphia, United States

**Roland Eils** Division of Theoretical Bioinformatics (B080), German Cancer Research Center (DKFZ), Heidelberg, Germany; Department for Bioinformatics and Functional Genomics, Institute for Pharmacy and Molecular Biotechnology (IPMB) and BioQuant, Heidelberg University, Heidelberg, Germany

**Wolfgang Enard** Department of Biology II, Ludwig Maximilian University Munich, Martinsried, Germany
https://orcid.org/0000-0002-4056-0550

**Andrew Farmer** Takara Bio United States, Inc., Mountain View, United States

**Lars Fugger** Oxford Centre for Neuroinflammation, Nuffield Department of Clinical Neurosciences, and MRC Human Immunology Unit, Weatherall Institute of Molecular Medicine, John Radcliffe Hospital, University of Oxford, Oxford, United Kingdom

**Berthold Göttgens** Department of Haematology, University of Cambridge, Cambridge, United Kingdom; Wellcome Trust-MRC Cambridge Stem Cell Institute, University of Cambridge, Cambridge, United Kingdom
https://orcid.org/0000-0001-6302-5705

**Nir Hacohen** Broad Institute of MIT and Harvard, Cambridge, United States; Massachusetts General Hospital Cancer Center, Boston, United States

**Muzlifah Haniffa** Institute of Cellular Medicine, Newcastle University, Newcastle upon Tyne, United Kingdom
https://orcid.org/0000-0002-3927-2084

**Martin Hemberg** Wellcome Trust Sanger Institute, Wellcome Genome Campus, Hinxton, United Kingdom

**Seung Kim** Departments of Developmental Biology and of Medicine, Stanford University School of Medicine, Stanford, United States

**Paul Klenerman** Peter Medawar Building for Pathogen Research and the Translational Gastroenterology Unit, Nuffield Department of Clinical Medicine, University of Oxford, Oxford, United Kingdom; Oxford NIHR Biomedical Research Centre, John Radcliffe Hospital, Oxford, United Kingdom

**Arnold Kriegstein** Eli and Edythe Broad Center of Regeneration Medicine and Stem Cell Research, University of California San Francisco, San Francisco, United States

**Ed Lein** Allen Institute for Brain Science, Seattle, United States
https://orcid.org/0000-0001-9012-6552

**Sten Linnarsson** Laboratory for Molecular Neurobiology, Department of Medical Biochemistry and Biophysics, Karolinska Institutet, Stockholm, Sweden

**Emma Lundberg** Science for Life Laboratory, School of Biotechnology, KTH Royal Institute of Technology, Stockholm, Sweden; Department of Genetics, Stanford University, Stanford, United States

**Joakim Lundeberg** Science for Life Laboratory, Department of Gene Technology, KTH Royal Institute of Technology, Stockholm, Sweden

**Partha Majumder** National Institute of Biomedical Genomics, Kalyani, India

**John C Marioni** Wellcome Trust Sanger Institute, Wellcome Genome Campus, Hinxton, United Kingdom; EMBL-European Bioinformatics Institute, Wellcome Genome Campus, Hinxton, United Kingdom; Cancer Research UK Cambridge Institute, University of Cambridge, Cambridge, United Kingdom

**Miriam Merad** Precision Immunology Institute, Icahn School of Medicine at Mount Sinai, New York, United States

**Musa Mhlanga** Division of Chemical, Systems & Synthetic Biology, Institute for Infectious Disease & Molecular Medicine (IDM), Department of Integrative Biomedical Sciences, Faculty of Health Sciences, University of Cape Town, Cape Town, South Africa

**Martijn Nawijn** Department of Pathology and Medical Biology, GRIAC Research Institute, University of Groningen, University Medical Center Groningen, Groningen, The Netherlands

**Mihai Netea** Department of Internal Medicine and Radboud Center for Infectious Diseases, Radboud University Medical Center, Nijmegen, The Netherlands

**Garry Nolan** Department of Microbiology and Immunology, Stanford University, Stanford, United States

**Dana Pe'er** Computational and Systems Biology Program, Sloan Kettering Institute, New York, United States

**Anthony Phillipakis** Broad Institute of MIT and Harvard, Cambridge, United States

**Chris P Ponting** MRC Human Genetics Unit, MRC Institute of Genetics & Molecular Medicine, University of Edinburgh, Edinburgh, United Kingdom
https://orcid.org/0000-0003-0202-7816

**Steve Quake** Department of Applied Physics and Department of Bioengineering, Stanford University, Stanford, United States; Chan Zuckerberg Biohub, San Francisco, United States

**Wolf Reik** Wellcome Trust Sanger Institute, Wellcome Genome Campus, Hinxton, United Kingdom; Epigenetics Programme, The Babraham Institute, Cambridge, United Kingdom; Centre for Trophoblast Research, University of Cambridge, Cambridge, United Kingdom

**Orit Rozenblatt-Rosen** Broad Institute of MIT and Harvard, Cambridge, United States

**Joshua Sanes** Center for Brain Science and Department of Molecular and Cellular Biology, Harvard University, Cambridge, United States
https://orcid.org/0000-0001-8926-8836

**Rahul Satija** Department of Biology, New York University, New York, United States; New York Genome Center, New York University, New York, United States
https://orcid.org/0000-0001-9448-8833

**Ton N Schumacher** Division of Immunology, The Netherlands Cancer Institute, Amsterdam, The Netherlands

**Alex Shalek** Broad Institute of MIT and Harvard, Cambridge, United States; Institute for Medical Engineering & Science (IMES) and Department of Chemistry, Massachusetts Institute of Technology, Cambridge, United States; Ragon Institute of MGH, MIT and Harvard, Cambridge, United States

**Ehud Shapiro** Department of Computer Science and Department of Biomolecular Sciences, Weizmann Institute of Science, Rehovot, Israel

**Padmanee Sharma** Department of Genitourinary Medical Oncology, Department of Immunology, MD Anderson Cancer Center, University of Texas, Houston, United States

**Jay W Shin** Division of Genomic Technologies, RIKEN Center for Life Science Technologies, Yokohama, Japan

**Oliver Stegle** EMBL-European Bioinformatics Institute, Wellcome Genome Campus, Hinxton, United Kingdom

**Michael Stratton** Wellcome Trust Sanger Institute, Wellcome Genome Campus, Hinxton, United Kingdom

**Michael JT Stubbington** Wellcome Trust Sanger Institute, Wellcome Genome Campus, Hinxton, United Kingdom

**Fabian J Theis** Institute of Computational Biology, German Research Center for Environmental Health, Helmholtz Center Munich, Neuherberg, Germany; Department of Mathematics, Technical University of Munich, Garching, Germany

**Matthias Uhlen** Science for Life Laboratory and Department of Proteomics, KTH Royal Institute of Technology, Stockholm, Sweden; Novo Nordisk Foundation Center for Biosustainability, Danish Technical University (DTU), Lyngby, Denmark

**Alexander van Oudenaarden** Hubrecht Institute and University Medical Center Utrecht, Utrecht, The Netherlands

**Allon Wagner** Department of Electrical Engineering and Computer Science and the Center for Computational Biology, University of California Berkeley, Berkeley, United States

**Fiona Watt** Centre for Stem Cells and Regenerative Medicine, King's College London, London, United Kingdom
  https://orcid.org/0000-0001-9151-5154

**Jonathan Weissman** Howard Hughes Medical Institute, Chevy Chase, United States; Department of Cellular & Molecular Pharmacology, University of California San Francisco, San Francisco, United States; California Institute for Quantitative Biomedical Research, University of California San Francisco, San Francisco, United States; Center for RNA Systems Biology, University of California San Francisco, San Francisco, United States
  https://orcid.org/0000-0003-2445-670X

**Barbara Wold** Division of Biology and Biological Engineering, California Institute of Technology, Pasadena, United States

**Ramnik Xavier** Broad Institute of MIT and Harvard, Cambridge, United States; Center for Computational and Integrative Biology, Massachusetts General Hospital, Boston, United States; Gastrointestinal Unit and Center for the Study of Inflammatory Bowel Disease, Massachusetts General Hospital, Boston, United States; Center for Microbiome Informatics and Therapeutics, Massachusetts Institute of Technology, Cambridge, United States

**Nir Yosef** Ragon Institute of MGH, MIT and Harvard, Cambridge, United States; Department of Electrical Engineering and Computer Science and the Center for Computational Biology, University of California Berkeley, Berkeley, United States
  https://orcid.org/0000-0001-9004-1225

**Human Cell Atlas Meeting Participants**

*Author contributions:* Aviv Regev, Sarah A Teichmann, Eric S Lander, Conceptualization, Writing—original draft; Ido Amit, Christophe Benoist, Ewan Birney, Bernd Bodenmiller, Peter Campbell, Piero Carninci, Menna Clatworthy, Hans Clevers, Bart Deplancke, Ian Dunham, James Eberwine, Roland Eils, Wolfgang Enard, Andrew Farmer, Lars Fugger, Berthold Göttgens, Nir Hacohen, Muzlifah Haniffa, Martin Hemberg, Seung Kim, Paul Klenerman, Arnold Kriegstein, Ed Lein, Sten Linnarsson, Emma Lundberg, Joakim Lundeberg, Partha Majumder, John C Marioni, Miriam Merad, Musa Mhlanga, Martijn Nawijn, Mihai Netea, Garry Nolan, Dana Pe'er, Anthony Phillipakis, Chris P Ponting, Stephen Quake, Wolf Reik, Orit Rozenblatt-Rosen, Joshua Sanes, Rahul Satija, Ton N Schumacher, Alex Shalek, Ehud Shapiro, Padmanee Sharma, Jay W Shin, Oliver Stegle, Michael Stratton, Michael J T Stubbington, Fabian J Theis, Matthias Uhlen, Alexander van Oudenaarden, Allon Wagner, Fiona Watt, Jonathan Weissman, Barbara Wold, Ramnik Xavier, Nir Yosef, Conceptualization, Writing—review and editing

*Competing interests:* Chris P Ponting: Reviewing Editor, *eLife*. Aviv Regev: Senior Editor, *eLife*. Fiona Watt: Deputy Editor, *eLife*. The other authors declare that no competing interests exist.

## Funding

The authors declare that there was no funding for this work

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
