## [Decision Letter]

Thank you for submitting your article "The human cell atlas" to *eLife* for consideration as a Feature Article. Your article has been favorably evaluated by a Senior Editor and three reviewers, one of whom is a member of our Board of Reviewing Editors. The reviewers have opted to remain anonymous.

The reviewers have discussed the reviews with one another and the Reviewing Editor has drafted this decision to help you prepare a revised submission.

Summary:

The reviewers were in agreement on the importance, timeliness, purpose and the potential value of the manuscript. For the most part the article is clearly written but there are several important issues that need clarification and several topics that needed to be included. In brief these are:

1) Definition: The manuscript does not do a convincing job of defining "cell type". This definition is essential for the reader to understand what the focus of the Atlas will be. The authors themselves point out that we lack an operational definition of "cell type". Single-cell genomics folks and others have grappled with defining cell type recently (see e.g. http://dx.doi.org/10.1016/j.cels.2017.03.006). However, an operational definition would be helpful for the wider community.

2) Single Cell Focus: It is made clear in the manuscript that use of single cell (SC) analysis is considered a cornerstone of the project but the text reads very much like an advertisement for SC technology. Although this approach will be needed and helpful for multiple reasons, there is a lack of balance in describing and justifying the project goal for finding, defining, characterizing and cataloguing as many cell types as possible vs. highlighting the single cell technology itself. Also of concern, is that while issues associated with SC studies are mentioned, based on the current status of the technology, it is not clear that the technology is currently ready to address issues of single cell epigenome sequencing (ChIP-seq, Hi-C, ATAC-seq). The manuscript would benefit by re-enforcing the main goal of the project of defining and cataloguing a human cell atlas rather than focusing on the single cell techniques and issues. Finally, by shifting the emphasis in this direction, this would offer the opportunity to broaden the audience for the manuscript to areas of cell sciences outside of genomics and make these audiences feel that there are areas for them to contribute to.

3) Samples: The origin of human material for cell type characterization and issues related to heterogeneity due to variable processing and inter-patient heterogeneity is not sufficiently discussed. The availability and uniform processing of human materials are critical for comprehensive and reproducible characterization of human cell types. Will cell type characterization be done mainly on post-mortem samples and if so, it would be important to mention that there will be different pipelines for their isolation and preparation (e.g. brain vs. other organs/tissues). Will other sources of samples be used such as: biopsies, material from surgical organ resections, human cell culture, or xenograft models? While samples of all these kinds will likely be utilized, the availability and pros and cons of the various means of sample selection and generation should be discussed. Which strategies for standardizing pipelines of sample collection and processing across different laboratories are envisioned?

4) Scope: The authors describe parallels between the Human Genome Project and the HCA, and highlight a few differences. But the biggest difference between the two is that the HGP had a fairly clear scope, or at least a clear first stopping point: a complete description of the human genome. Fully interpreting that sequence is a vast undertaking that was seen as beyond the scope of the HGP. The authors should clearly define the minimal deliverables of HCA version 1.0, or at least list a menu of deliverables to drive the discussion about what HCA version 1.0 should include. This could be presented as the first figure or table.

5) Cost-Benefit Issue: There is no discussion of the opportunity cost of the HCA. There will be two audiences reading this paper. The first will be our scientific colleagues that will ask the traditional question of "What does this have to do and how will it affect my work?". With this comes the danger that a diverse scientific audience will not be persuaded that there is a place for them with this project and that funds ear-marked for the HCA project will reduce support for smaller efforts. Readers might wonder whether it might not be better to just keep funding individual investigators and projects within that community, rather than (presumably) a massive, hierarchical, consortia-driven project. The manuscript should dedicate some space to discussing why the HCA is more deserving than the numerous other smaller projects that won't be funded. A second audience will be non-scientific. This audience will bring many social, economic, ethical perspectives. To address this audience might well require a different venue (i.e. a news and views-like article). However, since many publication outlets (journals, newspapers, etc.) will have writers reporting on this initial publication, it would be worth summarizing the benefits, the likely financial costs and timelines in a manner that is clear and does not need distillation of the paper's contents. This may be of importance given today's political environment since it may lower (not eliminates) possible mis-interpretation (intentional and unintentional) of these topics by non-scientific readers.

6) Model Organisms: The topic of the role of model organisms in the project is mentioned. The manuscript notes that Atlases for model organisms will be essential because we can probe the link between molecular signature and function. The authors should discuss more fully the possibility of creating an atlas for one model organism before attempting the human atlas. This strategy would provide a way to discover if meaningful molecular signatures are being collected, evaluate different sample collection and preparation strategies and provide benchmarks for different technologies that will be used. Alternatively, is it the intention of the consortium to use one or more model organism only as a system for specific functional follow-up experiments concerning individual cell types? If this is the case how many model organisms are envisioned to be used and what organ systems in each model organism will be studied?

---

## [Author Response]

Summary:The reviewers were in agreement on the importance, timeliness, purpose and the potential value of the manuscript. For the most part the article is clearly written but there are several important issues that need clarification and several topics that needed to be included. In brief these are:1) Definition: The manuscript does not do a convincing job of defining "cell type". This definition is essential for the reader to understand what the focus of the Atlas will be. The authors themselves point out that we lack an operational definition of "cell type". Single-cell genomics folks and others have grappled with defining cell type recently (see e.g. http://dx.doi.org/10.1016/j.cels.2017.03.006). However, an operational definition would be helpful for the wider community.

In fact, this choice is *deliberate*. We strongly believe that it is premature, at this time, to settle on a fixed definition of “cell type.” Rather, the *right* definition of “cell type” will need to emerge from data (just as the definition of “gene” became clear only after looking at a large number of genes). Indeed, a recent article in Cell Systems (2017) makes clear that experts in the field currently have quite different notions of “cell type” (and many experts who were invited to contribute to the article, including several of the authors of this manuscript, declined to do so because the notion is still in flux).

To address the reviewer’s concerns, we have modified the text to clarify this issue (subsection “Taxonomy: Cell types”, second and third paragraphs).

2) Single Cell Focus: It is made clear in the manuscript that use of single cell (SC) analysis is considered a cornerstone of the project but the text reads very much like an advertisement for SC technology. Although this approach will be needed and helpful for multiple reasons, there is a lack of balance in describing and justifying the project goal for finding, defining, characterizing and cataloguing as many cell types as possible vs. highlighting the single cell technology itself. Also of concern, is that while issues associated with SC studies are mentioned, based on the current status of the technology, it is not clear that the technology is currently ready to address issues of single cell epigenome sequencing (ChIP-seq, Hi-C, ATAC-seq). The manuscript would benefit by re-enforcing the main goal of the project of defining and cataloguing a human cell atlas rather than focusing on the single cell techniques and issues. Finally, by shifting the emphasis in this direction, this would offer the opportunity to broaden the audience for the manuscript to areas of cell sciences outside of genomics and make these audiences feel that there are areas for them to contribute to.

We believe that the description of the technology is not excessive – it consumes only 7 paragraphs (out of 93) – and is essential to making the case that it is feasible now to start a Human Cell Atlas Project. These paragraphs cover a range of approaches, including single cell RNA-Seq, genomics and epigenomics, and spatial methods.

To address the reviewer’s concerns, we have modified the text to (i) clarify that the HCA is an *intellectual* endeavor, for which the technologies will evolve over time and (ii) note that some technologies (e.g., for epigenomic characterization) are not as mature as others.

To address the reviewer’s concerns, we have modified the text to clarify this issue (Introduction, last paragraph).

3) Samples: The origin of human material for cell type characterization and issues related to heterogeneity due to variable processing and inter-patient heterogeneity is not sufficiently discussed. The availability and uniform processing of human materials are critical for comprehensive and reproducible characterization of human cell types. Will cell type characterization be done mainly on post-mortem samples and if so, it would be important to mention that there will be different pipelines for their isolation and preparation (e.g. brain vs. other organs/tissues). Will other sources of samples be used such as: biopsies, material from surgical organ resections, human cell culture, or xenograft models? While samples of all these kinds will likely be utilized, the availability and pros and cons of the various means of sample selection and generation should be discussed. Which strategies for standardizing pipelines of sample collection and processing across different laboratories are envisioned?

As noted above, the purpose of this paper is to describe the overall *concept* of the Human Cell Atlas. Technical aspects about the origins, sources, handling, and processing of human samples are discussed at length in Section 2 of the HCA White paper.

To address the reviewer’s concerns, we have modified the text to make this clear by pointing to the White Paper and summarizing key salient points (subsection “Toward a Human Cell Atlas”).

4) Scope: The authors describe parallels between the Human Genome Project and the HCA, and highlight a few differences. But the biggest difference between the two is that the HGP had a fairly clear scope, or at least a clear first stopping point: a complete description of the human genome. Fully interpreting that sequence is a vast undertaking that was seen as beyond the scope of the HGP. The authors should clearly define the minimal deliverables of HCA version 1.0, or at least list a menu of deliverables to drive the discussion about what HCA version 1.0 should include. This could be presented as the first figure or table.

While the HCA White Paper contains a lengthy definition of the intended scope, we agree that this paper would benefit by expanding the description of the scope.

We have therefore expanded the description and also pointed to Sections 1 and 2 of the HCA White Paper for the detailed description of the scope of the project, including its first draft, full version, and bounds (subsection “Toward a Human Cell Atlas”).

5) Cost-Benefit Issue: There is no discussion of the opportunity cost of the HCA. There will be two audiences reading this paper. The first will be our scientific colleagues that will ask the traditional question of "What does this have to do and how will it affect my work?". With this comes the danger that a diverse scientific audience will not be persuaded that there is a place for them with this project and that funds ear-marked for the HCA project will reduce support for smaller efforts. Readers might wonder whether it might not be better to just keep funding individual investigators and projects within that community, rather than (presumably) a massive, hierarchical, consortia-driven project. The manuscript should dedicate some space to discussing why the HCA is more deserving than the numerous other smaller projects that won't be funded. A second audience will be non-scientific. This audience will bring many social, economic, ethical perspectives. To address this audience might well require a different venue (i.e. a news and views-like article). However, since many publication outlets (journals, newspapers, etc.) will have writers reporting on this initial publication, it would be worth summarizing the benefits, the likely financial costs and timelines in a manner that is clear and does not need distillation of the paper's contents. This may be of importance given today's political environment since it may lower (not eliminates) possible mis-interpretation (intentional and unintentional) of these topics by non-scientific readers.

We agree with the reviewers on both points.

For the scientific community, we have added a subsection (“(9) Impact on the scientific community”) addressing the benefits and potential costs to the broad scientific community.

For the non-scientific community, we agree with the reviewers that the main format for this interaction is through other means than this manuscript. The Organizing Committee of the HCA (sharing many co-authors of this manuscript) has published a Commentary recently (Nature, October 2017) as a concise summary of the HCA white paper. As described in the white paper (Section 7) we are committed to diverse forms of outreach to the broad non-scientific community. We briefly discuss this in the revised manuscript (subsection “(10) Engagement with the non-scientific community”).

6) Model Organisms: The topic of the role of model organisms in the project is mentioned. The manuscript notes that Atlases for model organisms will be essential because we can probe the link between molecular signature and function. The authors should discuss more fully the possibility of creating an atlas for one model organism before attempting the human atlas. This strategy would provide a way to discover if meaningful molecular signatures are being collected, evaluate different sample collection and preparation strategies and provide benchmarks for different technologies that will be used. Alternatively, is it the intention of the consortium to use one or more model organism only as a system for specific functional follow-up experiments concerning individual cell types? If this is the case how many model organisms are envisioned to be used and what organ systems in each model organism will be studied?

We agree that additional information about the considerations around model organism atlases is important. This subject is very much within the scope of the initiative, and we have added a paragraph (subsection “(5) Model organisms”) to discuss it.

We note that model organisms will only be covered in detail in the White Paper following a workshop (to be held during November 2017 in Berlin) devoted to this subject.